# Efficient Lithium-Based Upconversion Nanoparticles for Single-Particle Imaging and Temperature Sensing

**DOI:** 10.3390/ma16124354

**Published:** 2023-06-13

**Authors:** Yahya A. Alzahrani, Abdulaziz Alromaeh, Masfer Alkahtani

**Affiliations:** 1Future Energy Technologies Institute, King Abdulaziz City for Science and Technology (KACST), Riyadh 11442, Saudi Arabia; yalzhrani@kacst.edu.sa; 2Microelectronics and Semiconductors Institute, King Abdulaziz City for Science and Technology (KACST), Riyadh 11442, Saudi Arabia

**Keywords:** upconversion nanoparticles, temperature, single-particle imaging, quantum sensing

## Abstract

Upconversion Nanoparticles (UCNPs) have attracted exceptional attention due to their great potential in high-contrast, free-background biofluorescence deep tissue imaging and quantum sensing. Most of these interesting studies have been performed using an ensemble of UCNPs as fluorescent probes in bioapplications. Here, we report a synthesis of small and efficient YLiF_4_:Yb,Er UCNPs for single-particle imaging as well as sensitive optical temperature sensing. The reported particles demonstrated a bright and photostable upconversion emission at a single particle level under a low laser intensity excitation of 20 W/cm^2^. Furthermore, the synthesized UCNPs were tested and compared to the commonly used two-photon excitation QDs and organic dyes and showed a nine times better performance at a single particle level under the same experimental conditions. In addition, the synthesized UCNPs demonstrated sensitive optical temperature sensing at a single particle level within the biological temperature range. The good optical properties of single YLiF_4_:Yb,Er UCNPs open an avenue for small and efficient fluorescent markers in imaging and sensing applications.

## 1. Introduction

Fluorescent nanomaterials have been grown or synthesized for their use as fluorescent markers or probes in quantum sensing and super-resolution imaging in many promising applications [1,2,3]. Most conventional fluorescent probe systems, such as organic dyes and quantum dots (QDs), exhibit a strong light emission in the visible range via the optical downconversion process when excited by short wavelengths (ultraviolet-blue) [4,5]. However, the use of short wavelength excitations leads to unwanted absorption by surrounding biological species, resulting in local overheating, photobleaching and auto-fluorescence background [6,7]. Upconversion organic dyes have been proposed as good alternative fluorescent markers, but they require a high laser excitation threshold and suffer from fast photobleaching, which is not preferable for most biological applications [8]. In addition, semiconductor quantum dots (QDs) are bright and stable fluorescent markers [9], but they display photoblinking at a small particle size of less than 10 nm and exhibit toxicity concerns [10].

To overcome these drawbacks, lanthanide-ion-doped (rare-earth) upconversion nanomaterials (UCNPs) are of special interest due to their tunable, photostable and sharp optical emissions upon low near-infrared (NIR) laser intensity excitations [11,12,13]. UCNPs also display a stable and complete absence of auto-fluorescence background, which improves the signal-to-noise ratio, enabling sensitive bio-sensing and high-contrast imaging [13,14]. The weak quantum yield of UCNPs is the only obstacle for their application at a single particle level. To solve this issue and successfully make a full use of the unique advantages of UCNPs, tremendous theoretical studies and experiments have been conducted to protect the surface of UCNP crystals with the addition of inert shells in order to prevent the surface-quenching effects caused by the surrounding species such as surface defects, optical traps and solvent ligands [15,16,17,18]. As a result, the best brightness of UCNPs during single-particle imaging at sub-10 Wcm^−2^ was optimized by carefully adjusting sensitizer and activator ion concentrations. These successful optimization steps result in a relatively high quantum yield reaching 5% under low laser irradiance [16,17,19]. The core–shell architecture engineering of UCNPs enabled useful applications of sodium-fluoride-based UCNPs in many promising applications.

Recently, special attention has been focused on the synthesis of lanthanide-ion-doped lithium-fluoride (LiYF_4_:Yb,Er) UCNPs for quantum sensing and laser refiguration applications. In the composition of these UCNPs, an optical upconversion process takes place inside a lithium yttrium fluoride host crystal doped with rare-earth ions (Yb and Er ions). In the upconversion process, Yb functions as a sensitizer ion with large absorption cross-sections at near-infrared (NIR) wavelengths and sequentially transfers absorbed energy to upconverting ions (Er) to emit visible light upon NIR excitations. It was shown that cubic lithium yttrium fluoride (Yb-doped LiYF_4_) can offset the photothermal heating of nanodiamonds by the anti-Stokes fluorescence cooling of Yb^3+^ ions under a 1030 nm laser excitation [20,21]. Thus, such an important application requires a careful optical performance assessment of lithium-based UCNPs at a single particle level.

In this work, we report a successful and simple synthesis of small LiYF_4_:Yb,Er UCNPs with an average size of 18 nm. The optical properties of the prepared UCNPs were investigated under the relatively low power intensity of an NIR laser and then tested for single-particle imaging. Compared to organic dyes and QDs, the synthesized UCNPs showed a relatively higher and stable single-particle emission at the same excitation intensity. In addition, the synthesized UCNPs showed sensitive optical temperature sensing, which makes them of special interest for many promising applications.

## 2. Materials and Methods

### 2.1. Materials

Ytterbium (III) chloride hexahydrate, yttrium (III) chloride hexahydrate and erbium (III) chloride hexahydrate are lanthanide elements (rare-earth chlorides). Lithium hydroxide, ammonium fluoride, oleic acid, 1-octadecene, methanol, ethanol, cyclohexane and deionized water were purchased from Sigma-Aldrich without any further purifications.

### 2.2. Synthesis of LiYF4:Yb,Er (18/2%) Upconversion Nanocrystals

An amount of 1.0 mmol of LnCl_3_.6H_2_O (80% of Y, which is equivalent to 0.8 mmole; 18% of Yb, which is equivalent to 0.18 mmole; and 2% of Er, which is equivalent to 0.02 mmole) was dissolved in 10.5 mL of oleic acid and 10.5 mL of 1-octadecene in a two-neck chemistry flask. This prepared mixture was gradually heated to 160 °C for 40 min under argon flow to form a transparent solution. After this step was complete, the solution was cooled down to 60 °C, and 5 mL of 2.5 mmol LiOH.H_2_O and 10 mL of 4.0 mmol NH_4_F were dissolved in methanol, vortexed for 10 s and then quickly injected into the reaction flask. The whole reaction was kept at 50 °C for 40 min with vigorous stirring to complete the required nucleation. Before increasing the synthesis temperature, it was necessary to remove access to methanol and water. To do this, the reaction temperature was raised to 160 °C for 20 min. In order to obtain a well-crystallized particles, the mixture was then heated to 300 °C under argon flow for 1.5 h. Finally, the synthesis temperature was reduced to room temperature, and the obtained LiYF_4_:Yb,Er UCNPs were collected after appropriate washing with ethanol. The washing procedure was repeated three times, and the synthesized UCNPs were re-dispersed in 10 mL of chloroform. It is highly recommended to store the final product at 4 °C for further use.

### 2.3. Ligand-Surface Exchange of LiYF4:Yb,Er UCNPs

The synthesized oleate-capped LiYF4:Yb,Er UCNPs (1.0 mL) were dispersed in a volume of 30 mL of acidic ethanol solution (pH = 1). The acidity of the ethanol solution was adjusted by concentrated hydrochloric acid [22]. A strong sonication was applied to the solution for 1 h to remove the oleate ligands and allow hydroxide ligand formation on the UCNPs’ surface. The water-soluble UCNPs were collected after appropriate centrifugation at 14,000 rpm for 20 min and after being washed several times with ethanol/water (1:1 *v*/*v*). The water-soluble UCNPs were then re-dispersed in DI water for further use.

### 2.4. Transmission Electron Microscope Imaging

TEM micrographs of the nanoparticles were obtained using a Titan 300 kV ST (FEI) electron microscope. Prior to TEM imaging, we treated the TEM grids with plasma to remove organics and dust and make the TEM grids hydrophilic for sample solutions. For TEM imaging, a few drops of samples were drop-casted on the TEM grids and dried under vacuum.

### 2.5. Steady-State Absorption

The absorption spectrum was recorded using a PerkinElmer Lambda 950 UV-Vis-NIR spectrometer. The synthesized UCNPs were diluted 10 times and placed in a quartz cuvette in the spectrometer machine, and the absorption spectrum was scanned over a wide range of wavelength from 300 nm to 1200 nm.

### 2.6. Upconversion Emission Measurements

Upconversion emission of the sample was characterized using a wavelength-tunable femtosecond pulsed Ti:sapphire laser (Coherent, Chameleon Ultra) with a pulse width = 140 fs and repetition rate = 80 MHz with an excitation light source at an excitation wavelength of 980 nm. The laser output beam was expanded by a 5× beam expander (Thorlabs Inc., Newton, MA, USA, GBE05-B) and attenuated by a variable iris diaphragm to approximately 0.5 cm in diameter. The intensity of the laser was controlled by rotating a half-wave plate (Thorlabs Inc., Newton, MA, USA; AHWP10M-980) mounted at the front of a laser-Glan polarizer (Thorlabs Inc., Newton, MA, USA, GL10-B). Linearly polarized light was used in the experiment. The intensity of the beam was monitored by a digital power meter (Thorlabs Inc., Newton, MA, USA, PM100D) equipped with thermal power sensors (Thorlabs Inc., Newton, MA, USA, S470C). The excitation laser beam was focused with an achromatic cylindrical lens (50 mm focal length) onto a 2 mm thick quartz cuvette (Thorlabs Inc., Newton, MA, USA, CV10Q700) mounted in a cuvette holder (Thorlabs Inc., Newton, MA, USA, CVH100) and equipped with a collimating optic and fiber adapter. The fluorescence was collected at an angle perpendicular to the direction of the excitation beam using a multimode optical fiber (Thorlabs Inc., Newton, MA, USA, BFL200LS02), which was connected to a spectrograph (Andor Technology, Newton, MA, USA, SR-303i-B) equipped with an EMCCD camera (Andor Technology, Newton, MA, USA, 970P-BV).

### 2.7. Downconversion Emission

Downconverted emission spectra were recorded using a spectrograph (Princeton Instruments, NJ, USA, Isoplane SCT320) equipped with a cooled IR camera (Princeton Instruments, NJ, USA). The UCNP samples were placed into the cuvette holder (Thorlabs Inc., Newton, MA, USA, CVH100) equipped with a 1050 nm long-pass edge filter (Thorlabs Inc., Newton, MA, USA, FELH1050) and an optical collimator and then connected to the spectrograph. Samples were excited using a Ti:sapphire 980 nm laser (Spectra-Physics, Santa Clara, CA, USA, Mai Tai) with a pulse width = 100 fs and repetition rate = 80 MHz. The laser output beam was expanded with a 5× beam expander (Thorlabs Inc., Newton, MA, USA, GBE05-B) and attenuated by a variable iris diaphragm at approximately 0.5 cm in diameter. The intensity of the laser beam was controlled by rotating a half-wave plate (Thorlabs Inc., Newton, MA, USA; AHWP10M-980) mounted at the front of a laser-Glan polarizer (Thorlabs Inc., Newton, MA, USA, GL10-B) and attenuated to a 135 mW average power. The intensity of the beam was monitored by a digital power meter (Thorlabs Inc., Newton, MA, USA, PM100D) equipped with thermal power sensors (Thorlabs Inc., Newton, MA, USA, S470C).

### 2.8. Single-Particle Confocal Microscopy (Upconversion/Multi-Photon Excitation)

Single-particle imaging measurements were performed on a home-built scanning confocal microscope (Olympus, Tokyo, Japan, IX 71) setup equipped with a Ti:sapphire 980 nm laser as an excitation source. Samples were excited using a semi apochromat water dipping objective (Olympus, Tokyo, Japan, LUMFL N, 60×/1.10 W). The optical scan of the samples was performed using raster scanning by moving the objective with three-axis piezoelectric translation stage (Physik Instrumente, Karlsruhe, Germany, P-733.2CL). The fluorescence of the samples was collected by the same objective, passed through a short-pass dichroic filter (Semrock, Rochester, NY, USA, FF750-SDi02), and filtered from the excitation beam by a short-pass filter (Thorlabs Inc., Newton, MA, USA, FESH0700). During measurements of single-particle luminescence images, the luminescence signal of each nanoparticle was detected using a single photon counter (Thorlabs Inc., Newton, MA, USA, SPAD, PicoQuant, τ-SPAD-50). The samples were excited by a 980 nm laser beam cleaned up by a laser line filter (Thorlabs Inc., Newton, MA, USA, FB980-10). The intensity of the excitation beam was monitored using a digital power meter (Thorlabs Inc., Newton, MA, USA, PM100D) equipped with a microscope slide thermal sensor (Thorlabs Inc., Newton, MA, USA, S175C) and attenuated to 2 mW at the output of the objective lens. Luminescence intensity distribution images were acquired with 200 *×* 200 pixel resolution and 2.5 ms accumulation time at each pixel in monodirectional scanning mode. Two-photon-induced luminescence of quantum dots (Invitrogen, OH, USA, Qdot 545 ITK carboxyl quantum dots) and 0.051 µm Dragon Green polystyrene beads (Bang Labs, OH, USA, FS02F) under the same experimental conditions were used as references to compare luminescence brightness.

### 2.9. Single-Particle Confocal Microscopy (Downconversion)

Measurement was carried out using NIR (980 nm) pulse excitation with average power of 7.2 mW. The excitation beam was filtered through a band pass filter (Thorlabs Inc., Newton, MA, USA, FBFB950-10). The confocal microcopy setup (as described above) was equipped with a dichroic beam splitter (Thorlabs Inc., Newton, MA, USA, DMLP1000R) and a long-pass emission filter (Thorlabs Inc., Newton, MA, USA, FELH1200). The photons emitted by the sample were detected using an InGaAs photodiode (Thorlabs Inc., Newton, MA, USA, MPD InGaAs) with 25 ms integration time per pixel to give a 200 *×* 200 pixel size map of the emission intensity distribution.

## 3. Results and Discussion

Experimentally, LiYF_4_:Yb,Er UCNPs were synthesized following a partially modified hydrothermal synthesis procedure previously reported in [23] and detailed in Materials and Methods. The structural characterizations (morphology and size) of the synthesized UCNPs were performed using a transmission electron microscope (TEM) and dynamic light scattering (DLS). For this, a TEM grid was prepared by dropping a few drops of diluted UCNPs (10×) in cyclohexane. Figure 1a shows a TEM image of the dispersed and tetragonal structure of the synthesized LiYF_4_:Yb,Er UCNPs. A high-resolution TEM image also illustrates well-crystallized UCNPs with an average size of 18 nm, as illustrated in Figure 1b. The narrow particle size of the synthesized UCNPs was also confirmed to be around 18 nm using dynamic light scattering (DLS).

The near-infrared absorption of the synthesized UCNPs was recorded using a UV-VIS-IR spectrophotometer. Figure 1d shows a clear NIR absorption band at 980 nm, which corresponds to the Yb^3+^ and Er^3+^ absorption bands. Among the rare-earth ions, an erbium (Er^3+^) ion was chosen as an upconverting ion in the UCNP composite as it has green and red emissions located within the first biological transparency window. Ytterbium (Yb) was also introduced into the UCNP composition as it has a large near-infrared absorption cross-section, which is better than that of Er^3+^, and functions as a photon (energy transfer) sensitizer to enhance the upconversion process in Er^3+^.

The uipconversion emission of the sample was characterized using a wavelength-tunable femtosecond pulsed Ti:sapphire laser (Coherent, Santa Clara, CA, USA, Chameleon Ultra) with a pulse width = 140 fs and repetition rate = 80 MHz, with an excitation light source at an excitation wavelength at 980 nm. The fluorescence was collected at an angle perpendicular to the direction of the excitation beam using a multimode optical fiber (Thorlabs, BFL200LS02), which is connected to a spectrograph (Andor Technology, Newton, MA, USA, SR-303i-B) equipped with an EMCCD camera (Andor Technology, Newton, MA, USA, 970P-BV). Details about the optical setup, which was used for up- and downconversion emission measurements, single-particle imaging and temperature sensing, can be found in Materials and Methods. Figure 2a shows an intensive visible (green and weak red) upconversion luminescence, which can be clearly observed by the naked eye under 980 nm at a laser intensity of 20 W/cm^2^. The observed optical emission spectrum revealed a strong and sharp emission band centered around 527–553 nm corresponding to the optical transitions (^2^H_11/2_ and ^4^S_3/2_ excited levels to ^4^I_15/2_ ground state) of the Er^3+^ ion as well as a weaker emission in the red region around 650 nm corresponding to another optical transition from ^4^F_9/2_ excited state to ^4^I_15/2_ ground states. In addition, a direct luminescence (downconversion) of Er^3+^ at 1.5 μm was recorded under the same NIR laser excitation (at 980 nm), which corresponds to the ^4^I_13/2_ to ^4^I_15/2_ transition, as depicted in Figure 2b. Interestingly, Figure 2c demonstrates the excellent photostability of the UCL of the synthesized UCNPs under a continuous wave (CW) of a 980 nm laser excitation over 1000 s.

The physics behind the up/downconversion luminescence observed from the synthesized UCNPs can be explained with Figure 2d. The optical luminescence in the UCNPs is a result of multiple and sequenced energy transfers between the sensitizer (Yb^3+^) and the activator (Er^3+^). The upconversion energy transfer mechanism can be explained with Figure 2d. First, Yb^3+^ which is well known to have a large absorption cross-section at 950–1000 nm, absorbs the first NIR photon and is excited to its ^2^F_5/2_ excited state and then transfers energy from Yb^3+^( excited state) to promote Er^3+^ to the semi-resonant metastable ^4^I_11/2_ level. As the semi-resonant metastable (^4^I_11/2_) level in the Er^3+^ has a long life time in milliseconds, a second NIR photon from the Yb^3+^ will promote Er^3+^ to higher excited states. After multiple nonradiative relaxations, two strong and sharp emission lines centered at 527 nm, 553 nm, and 650 nm will be emitted, respectively, from the ^2^H_11/2_, ^4^S_3/2_ and ^4^F_9/2_ excited states of Er^3+^ to ^4^I_15/2_ ground state. The lifetime quasi-resonant metastable level (^4^I_11/2_) in the Er^3+^ is long and at least long enough for upconversion to occur because the downconversion emission, compared to the upconversion emission, has a weak contribution of nonradiative relaxation processes from the ^4^I_11/2_ state to the ^4^I_13/2_ state [24]. The quantum yield (Q.Y) of the synthesized YLiF_4_:Yb,Er UCNPs was measured in this study. Compared to an infrared dye reference (IR-980), it was found that the Q.Y value for YLiF_4_:Yb,Er UCNPs with an average size of 18 nm is 0.18 ± 0.05% under a 980 nm laser excitation with a power intensity of 100 W·cm^−2^. This Q.Y. value is in good agreement with similar YLiF_4_:Yb,Er UCNPs reported in [25] and is a little bit higher than a similar NaYF4:Yb,Er core only reported in [15,16]. We expected a higher Q.Y. reaching 5% with appropriate core–shell–shell architectures similar to the higher values of YLiF_4_:Yb,Er UCNPs’ core–shell–shell particles reported in [16,25].

The single-particle imaging measurements in this study were performed on a home-built scanning confocal microscope (details about the optical setup can be found in Materials and Methods). For single-particle imaging measurements, the synthesized UCNPs were diluted 10 times and drop-casted on a quartz cover slip and then spin-coated to avoid unwanted agglomerations. The luminescence signal of each nanoparticle was detected using a single photon counter under a 980 nm laser excitation at an intensity of 20 W/cm^2^. Figure 3a shows a UC luminescence intensity image of the synthesized UCNPs distributed with 200 × 200 pixel resolution and a 2.5 ms accumulation time at each pixel in a monodirectional scanning mode. The downconversion emission of the synthesized UCNPs at single particle levels of the same partic spots was also acquired at the same excitation and imaging condition as shown in Figure 3b. It can be seen that the upconversion emission of the synthesized UCNPs is five times stronger than the 1.5 μm emission due to a faster nonradiative process in the upconversion regime.

It was important to compare the UC brightness of the synthesized UCNPs to conventional two-photon excitation fluorescent markers such as quantum dots and organic dyes at a single particle level and the same experimental conditions. Figure 3c,d show a direct comparison between UCNPs and two-photon-induced luminescence of quantum dots (Invitrogen, OH, USA, QDs 545 ITK carboxyl quantum dots) at the same excitation and imaging conditions. The average brightness of single UCNPs particles is almost eight times higher than single QDs.

Similarly, other single-particle brightness comparison measurements were carried out to compare the luminescence brightness of single UCNPs to Dragon Green polystyrene beads (Bang Labs, FS02F) under the same experimental conditions. Dragon Green polystyrene beads are commonly used in single-particle imaging in biology. Figure 4 shows that the brightness of single UCNPs is nine times higher than that of Dragon Green polystyrene beads. In comparison to QDs and polymeric NPs, UCNPs have a lower pump intensity threshold due to their long-lived metastable state (^4^I_11/2_) with a lifetime in the order of milliseconds. This exceptional optical property allows efficient single-particle imaging using UCNPS at a low power excitation compared to QDs and polymeric NPs, which are known to have higher quantum efficiency [26,27] at much higher laser intensity thresholds in the two-photon process. The higher pump threshold required for QDs and polymeric NPs adds more complications and results in unwanted damage and the overheating of the samples, especially in bioapplications. These single-particle imaging results indicate that the synthesized UCNPs are efficient and suitable for deep and autofluorescence-background-free imaging applications.

The optical temperature dependence of UC emission in YLF_4_:Yb, Er nanoparticles was investigated over a biocompatible temperature range. For this, UCNPs were spin-coated on a clean quartz substrate and placed in a temperature-controlled heating stage attached to the optical confocal microscope. Figure 5a shows the temperature dependence of the UC green emission over a temperature range of 298 K to 315 K under a low power 980 nm laser excitation at 20 W/cm^2^ to avoid any local heating, which could affect the optical temperature sensing measurements. UCNPs are known to sense temperature change dependence using Boltzmann populations in two thermally coupled excited state levels (^2^H_11/2_, ^4^S_3/2_) in erbium (Er^3+^) ions.

The fluorescence intensity ratio (*FIR*) of transitions from the ^2^H_11/2_ and ^4^S_3/2_ erbium excited states to the ^4^I_15/2_ erbium ground state as a function of temperature changes could be computed using Equation (1) [23], which is derived in detail in [28]:(1)$$FIR=\frac{I_{527}}{I_{552}}=Ce^{-\frac{\Delta E}{KT}}$$
where *C* is an empirical constant of the degeneracies of the erbium Er^3+^ (^2^H_11/2_, ^4^S_3/2_) excited states, emission cross-sections, and the fluorescence frequencies of the two green transitions; *K* is the Boltzmann’s constant; *T* is the temperature; and Δ*E* is the energy separation of two thermally coupled levels in the Er^3+^ excited states. Figure 5b shows a linear dependance between the *FIR* and temperature of the experimental data. The fitting slope between *FIR* and 1/*T* showed that energy difference between the thermally coupled excited states is 841 cm^−1^, which shows a correlation between theory and experiment.

Next, it was very important to evaluate the temperature sensitivity of the synthesized UCNPs over a range of temperature change. The relative sensor sensitivity (in % K^−1^) can be computed using Equation (2):(2)$$S=\frac{d\left(FIR\right)}{dT}=C\frac{\Delta E}{KT^{2}}e^{-\frac{\Delta E}{KT}}=100\%\left(\frac{\Delta E}{KT^{2}}\right)$$

Given that the calculated difference between the thermally coupled excited states of the Er^3+^ ion is 841 cm^−1^, a curve of the sensitivity as a function of temperature is plotted in Figure 5c. The synthesized UCNPs showed a very good temperature ratiometrically with sensitivities in the range of 1.2–1.3% K^−1^ within the biological temperature window, which is in good agreement with the high relative sensitivity of UCNPs reported in [7,28].

For more demanding applications, especially in biology, water-soluble particles are needed. To explore the water solubility of the synthesized oleate-capped particles without losing their good optical properties, a surface–ligand exchange experiment was carried out. For this, the synthesized oleate-capped UCNPs were dispersed in acidic ethanol with a pH equal to 1 under strong sonication for one hour and collected after appropriate washing steps [19,22]. Details on this surface–ligand exchange method can be found in Materials and Methods. Figure 6a shows a TEM image of well-dispersed water-soluble UCNPs without notable agglomerations. The water tolerant UCNPs retained their good optical luminescence in water with almost similar UC intensity compared to the oleate-capped UCNPs in an organic solvent under the same excitation conditions as illustrated in Figure 6b. In addition to the single-particle imaging applications of the synthesized UCNPs, we plan to grow diamond nanoshells around UCNPs to mitigate the current problems of background biofluorescence in bio-imaging applications with diamond color centers excited with visible light. Additionally, there is a great potential for Yb-doped YLF UCNPs shells coating nanodiamonds for laser refrigeration and optical traps in quantum applications.

## 4. Conclusions

In conclusion, small and efficient YLiF_4_:Yb,Er UCNPs were synthesized for single-particle imaging and optical temperature sensing. The structural characterizations of the synthesized UCNPs showed that they were well-dispersed, crystalline, and small-sized particles with a trigonal shape, which is common for lithium-based particles. Compared to conventional fluorescent dyes, organic dyes and QDs, the synthesized UCNPs showed a bright and photostable fluorescence under a low-power biocompatible NIR excitation at a single particle level. Furthermore, the synthesized UCNPs demonstrated a good temperature ratiometrically with high sensitivity within the biological temperature range. The synthesized oleate-capped were made water soluble by ligand–surface exchange and retained their optical properties under NIR excitation in a water solution. These good optical properties combined with efficient single-particle imaging make the YLiF_4_:Yb,Er UCNPs biocompatible and efficient fluorescent markers in many promising bioapplications.

## Figures and Tables

**Figure 1 materials-16-04354-f001:**
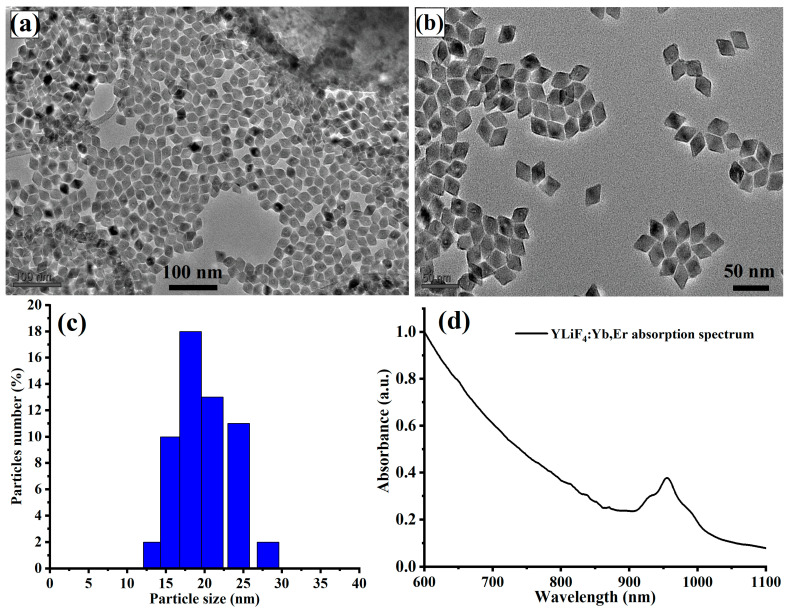
(**a**) Low-magnification TEM images of the synthesized YLiF_4_:Yb,Er UCNPs nanocrystals. (**b**) Magnified TEM image of the synthesized UCNPs. Both low- and high- magnification images of the UCNPs show small and well-dispersed particles with average size of 18 nm. (**c**) Particle size distribution of the synthesized UCNPs was recorded by dynamic light scattering (DLS), which shows a good agreement with particle sizes of the particles recorded by TEM. (**d**) The absorption spectrum was recorded using a PerkinElmer Lambda 950 UV-Vis-NIR spectrometer. The NIR absorption of the synthesized UCNPs peaked at 980 nm, which corresponds to the maximum absorption cross-section of Yb and Er rare-earth ions.

**Figure 2 materials-16-04354-f002:**
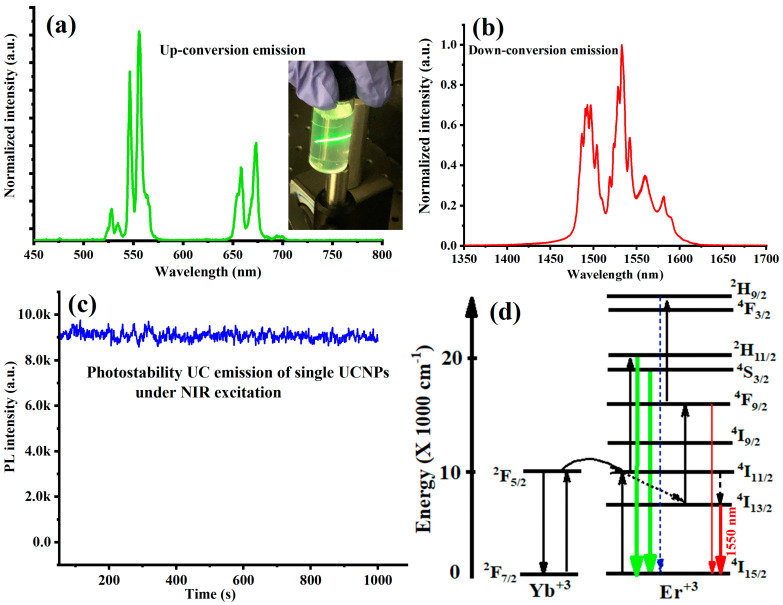
(**a**) Upconversion emission of the synthesized YLiF_4_:Yb,Er UCNPs deposited on a quartz substrate under a relatively low NIR laser excitation at intensity of 20 W/cm^2^. Inset: a photograph image of strong UC emission of UCNPs in solution. (**b**) Downconversion emission of the synthesized UCNPs peaked at 1.5 μm under a NIR laser excitation at intensity of 20 W/cm^2^ (**c**) Photostability of a single upconverting nanoparticle recorded after spin-coating the UCNPs on a quartz substrate under CW 980 nm excitation. (**d**) Schematic energy-level diagram of the Yb^3+^ and Er^3+^ systems showing upconversion energy transfer mechanism. Radiative processes and energy transfers are represented by full and dotted arrows, respectively.

**Figure 3 materials-16-04354-f003:**
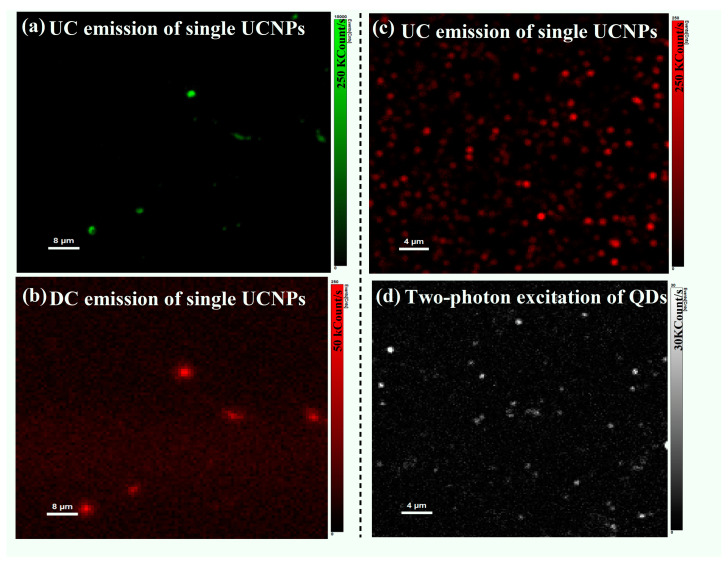
(**a**) A monodirectional optical scan of the synthesized UCNPs distributed with 200 × 200 pixel resolution and 2.5 ms accumulation time at each pixel. Upconversion emission from the optical scans shows a bright single upconverting particle with maximum UC intensity of 250 Kcounts/s. (**b**) Downconversion emission of the same scanned UCNPs with maximum DC intensity of 50 Kcounts/s. Both UC and DC optical characterizations of single UCNPs were performed at the same excitation and acquisition time conditions. (**c**,**d**) show a direct luminescence brightness comparison of single UC nanoparticles and two-photon excitation QDs under the same experimental conditions.

**Figure 4 materials-16-04354-f004:**
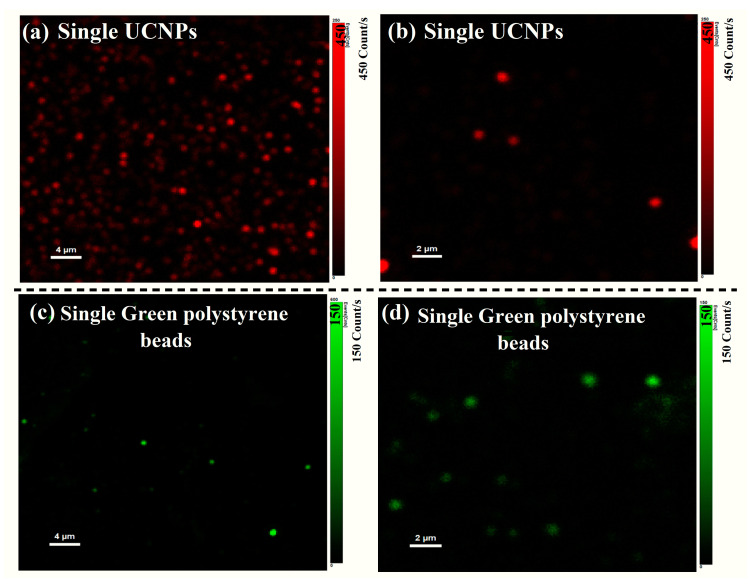
Optical scan of single UCNPs and single green polystyrene beads distributed with 200 × 200 pixel resolution and 2.5 ms accumulation time at each pixel. Upconversion emission from the optical scan in (**a**,**b**) show a bright single upconverting particle with maximum UC intensity of 450 Kcounts/s. (**c**,**d**) Optical scan of single green polystyrene beads show maximum two-photon emission intensity of 150 Kcounts/s.

**Figure 5 materials-16-04354-f005:**
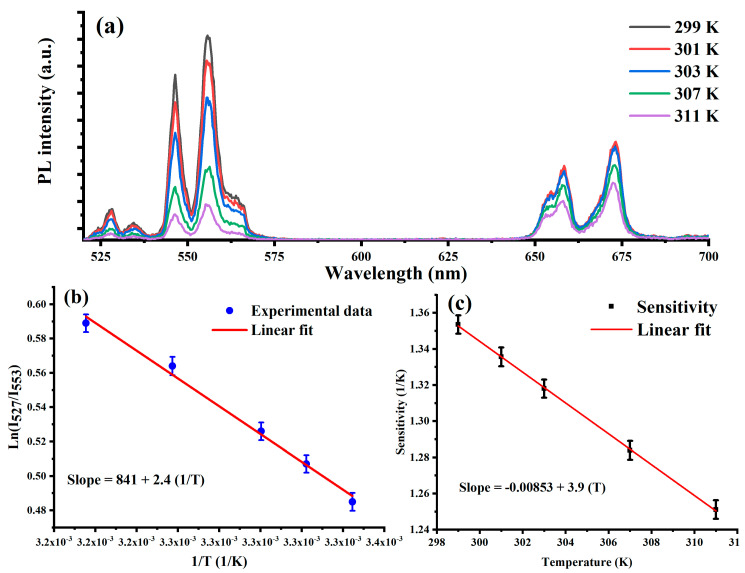
(**a**) Temperature dependence of upconversion luminescence emission spectrum of YLF4:Yb,Er UCNPs under 980 nm excitation recorded over a small temperature range (298–312 K). (**b**) linear fitting of the fluorescence intensity ratio (*FIR*) of the synthesized UCNPs as a function of temperature. (**c**) The recorded temperature sensitivity of the UCNPs over the biological temperature range of interest.

**Figure 6 materials-16-04354-f006:**
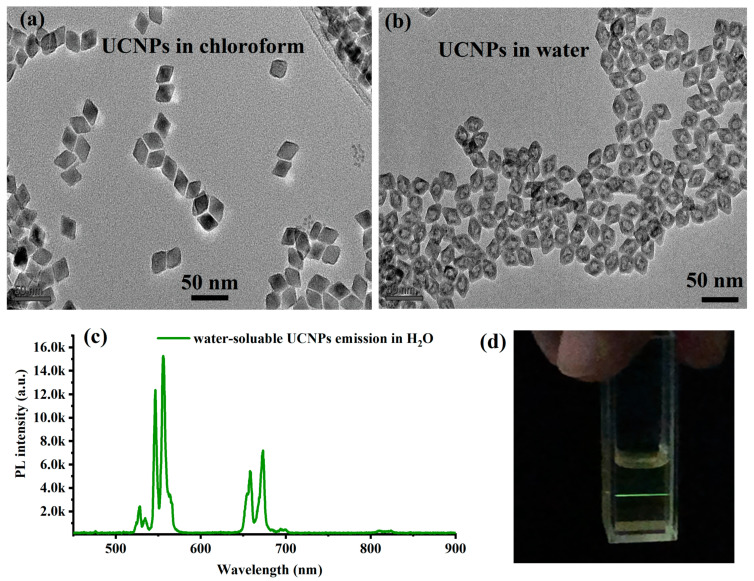
(**a**) TEM image of the oleate-capped UCNPs before ligand–surface exchange. (**b**) TEM image of the water-soluble UCNPs after ligand–surface exchange. (**c**) Upconversion emission from the water-soluble UCNPs under 980 nm laser excitation. (**d**) Optical image of the UC emission of the water-soluble UCNPs in water.

## Data Availability

The data presented in this study are available on request from the corresponding author.

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
