# Peer review of "Efficient Lithium-Based Upconversion Nanoparticles for Single-Particle Imaging and Temperature Sensing"

_materials, 2023, doi:10.3390/ma16124354_

Round 1

Reviewer 1 Report

The manuscript "Upconversion Nanoparticles for Single Particle Imaging and Quantum Sensing" is interesting for scientific community. I have just some suggestions for improving the work:

1. The title of the manuscript is very superficial. It needs to be something more specific. I suggest changing the title of the manuscript. (remember not to put the chemical elements with the chemical formulas, as they make it difficult to find these articles on scientific searchings).

2. Lines 55-61:  I did not understand what the authors would like to mention with this statement: "lithium fluoride based UCNPs as a bright and small particle with a trigonal shape"

3. I missed in section 2.2 an Scheme of experimental synthetic procedure. It would be very usefull for readers.

4. It is one curiosity: Why authors have treated with plasma Cu grids? 

5. Section 2.4 is poorly described. More experimental details must be written.

6. It was not clear in the manuscript the role of each component: YLiF4:Yb,Er. Explain in details in Introduction and Results and Discussion sections the main function of LiF4, Yb, and Eu in the Upconversion Nanoparticles.

Author Response

Referee 1’s comments

1) The manuscript "Upconversion Nanoparticles for Single Particle Imaging and Quantum Sensing" is interesting for scientific community. I have just some suggestions for improving the work:

Response: We thank the Referee for his/her valuable time for reviewing, kindly giving us constructive suggestions to improve our manuscript.

2) The title of the manuscript is very superficial. It needs to be something more specific. I suggest changing the title of the manuscript. (Remember not to put the chemical elements with the chemical formulas, as they make it difficult to find these articles on scientific searching).

Response:

We have changed the manuscript title to be more specific. The new title is “Efficient Lithium-based Upconversion Nanoparticles for Single Particle Imaging and Temperature Sensing” as shown in the revised manuscript.

3) Lines 55-61:  I did not understand what the authors would like to mention with this statement: "lithium fluoride based UCNPs as a bright and small particle with a trigonal shape"

Response:

We have changed the confusing statement to be clear and understandable. This change was highlighted in red and added into the revised manuscript in lines (58-59).

4) I missed in section 2.2 an Scheme of experimental synthetic procedure. It would be very useful for readers.

Response:

We thank the reviewer for his/her interesting suggestion, but we think that the text explanation of the synthetic procedure is enough, especially for the materials and methods section.

5) It is one curiosity: Why authors have treated with plasma Cu grids? 

Response:

We treat TEM grids with plasm prior to imaging to remove any organics, dust, and also to make the grids hydrophilic for sample solution. This is described in the TEM section in the manuscript in red in line (109-110).

6) Section 2.4 is poorly described. More experimental details must be written.

Response:

This section became section 2.5 in the revised manuscript. We have added enough experimental details as highlighted in red in lines (113-116).

7) It was not clear in the manuscript the role of each component: YLiF4:Yb,Er. Explain in details in Introduction and Results and Discussion sections the main function of LiF4, Yb, and Eu in the Upconversion Nanoparticles.

Response:

We really thank the reviewer for this helpful comment. The main functions of YLiF4:Yb,Er are explained in the introduction as highlighted in red the revised manuscript in lines (60-64). Also, it was discussed in details in the results and discussion as highlighted in red in lines (223-234).

*********************************************************************************************

Reviewer 2 Report

The MS materials-2453257 represents up-conversion lanthanide-doped lithium fluorides suitable for single particle imaging. The Introduction section claims two main problems in development of such NPs, which have not been solved yet. These problems are: low quantum yeild of the NPs and their coating, which is inconvenient for minimizing of radiationless losses. However, there is no comparative analysis of QYs of the synthesized NPs with similar up-conversion lanthanide fluorides or oxides. The comparison with the single examples of QDs and polymeric NPs is not enough for demonstrating a novelty of the represented NPs. Thus, comparative analysis of the QY values of the NPs with those reported in literature is required.

Unfortunatelly, authors pay very little attention on the coating of the NPs. From the Exp. Section one can find that the NPs are stabilized by oleates. Oleate-based coating is well-known. Commonly, oleate-stabilized nanostructures exhibit lower radiationless losses than those stabilized by hydrophilic coating. However, the former NPs are not suitable for bioimaging. But, may be an applicability of the single particle imaging is not limited by imaging, and there is no need to hydrophilize them? The future applicability of the developed NPs must be discussed.

Moderate style editing of the MS id required.

Author Response

Referee 2’s comment

The MS materials-2453257 represents up-conversion lanthanide-doped lithium fluorides suitable for single particle imaging. The Introduction section claims two main problems in development of such NPs, which have not been solved yet. These problems are: low quantum yeild of the NPs and their coating, which is inconvenient for minimizing of radiationless losses. However, there is no comparative analysis of QYs of the synthesized NPs with similar up-conversion lanthanide fluorides or oxides. The comparison with the single examples of QDs and polymeric NPs is not enough for demonstrating a novelty of the represented NPs. Thus, comparative analysis of the QY values of the NPs with those reported in literature is required.

Response:

First, we thank the Referee for his/her valuable time for reviewing, kindly giving us constructive suggestions to improve our manuscript.

Second, The quantum yield (Q.Y) of the synthesized YLiF4:Yb,Er UCNPs was measured in this study. Compared to an infrared dye reference (IR-980) as a reference, and found that the Q.Y value for YLiF4:Yb,Er UCNPs with average size 18 nm is 0.18 ± 0.05% under 980 nm laser excitation with power intensity (100 W·cm−2). This Q.Y. value is in a good agreement with similar YLiF4:Yb,Er UCNPs reported [25] in and a little bit higher than similar NaYF4:Yb,Er core only reported in[15, 16]. We expect a higher Q.Y. reaching 5% with appropriate core-shell-shell architectures similar to higher values of YLF4:Yb, Er core-shell-shell particles reported in[16, 25]. This paragraph was incorporated into the revised manuscript as highlighted in blue in lines (237-244). 

In addition, we compared the performance of synthesized UCNPs to both QDs and polymeric NPs at a single particle imaging in the following paragraph;

In comparison to QDs and polymeric NPs, UCNPs have a lower pump intensity threshold due to their long-lived metastable state (4I11/2) with a life time in order of milliseconds. This exceptional optical property allows an efficient single particle imaging using UCNPs at lower power excitation compared to QDs and polymeric NPs which are known to have higher quantum efficiency[26, 27] at higher laser intensity thresholds in two-photon process. The higher pump threshold required for QDs and polymeric NPs add more complications and results in unwanted damage and overheating of the samples, especially in bio-applications. This paragraph was incorporated into the revised manuscript as highlighted in blue in lines (288-295). 

Referee 2’s comment

Unfortunatelly, authors pay very little attention on the coating of the NPs. From the Exp. Section one can find that the NPs are stabilized by oleates. Oleate-based coating is well-known. Commonly, oleate-stabilized nanostructures exhibit lower radiationless losses than those stabilized by hydrophilic coating. However, the former NPs are not suitable for bioimaging. But, may be an applicability of the single particle imaging is not limited by imaging, and there is no need to hydrophilize them? The future applicability of the developed NPs must be discussed.

Response:

We thank the reviewer for his/her valuable comment.

For more demanding application, especially in biology, water-soluble particles are needed. To explore the water solubility of the synthesized oleate-capped particles without scarfing their good optical properties, surface–ligand exchange experiment was carried out. Figure 6(a) in the revised manuscript shows a TEM image of well dispersed water-soluble UCNPs without notable agglomerations. The water tolerant UCNPs retained their good optical luminescence in water with almost similar UC intensity compared to the oleate-capped UCNPs in organic solvent under the same excitation conditions as illustrated in Figure 6(b). Besides single particle imaging applications of the synthesized UCNPs, we plan to grow diamond nano shells around UCNPs to mitigate the current problems of background biofluorescence in bio-imaging applications with diamond color centers excited with visible light. Also, there is a great potential for Yb-doped YLF UCNPs shells coating nanodiamonds for laser refrigeration and optical traps in quantum applications. This paragraph was incorporated into the revised manuscript as highlighted in blue in lines (338-352). 

  • Figure 6 was added into the revised manuscript according to reviewer 2 comment.
  • Also, we have added a small section in the materials and methods section as (section 2.3).

Reviewer 3 Report

Reviewer

The manuscript titled “Upconversion Nanoparticles for Single Particle Imaging and Quantum Sensing” was revised for its possible publication in Materials MDPI.

I agree with the objectives, analyses, and techniques described by the authors after reading the complete manuscript. As a positive point, I found relevant the results presented. Nevertheless, the manuscript must improve some editing issues and spelling to comply with the quality standards of this journal.

Keywords. The first keyword is too long; one or up to two words for each keyword are acceptable.

Page 1. Merged brackets of references 1 to 14 need to be separated.

Orthography in Section 2.1. Reagents are incorrectly capitalized. Only the first word and after period should be in capital letters.

Section 2.4. Complete the information of this instrumental and measurement technique.

Figure 5 (a). Use a continuous line in stacked spectra because is difficult to distinguish overlapped intensities.

Figure 5 (b) and (c). Include error bars for each data point and add slope in Figure 5 (c).

Acknowledgments have a misspelling, correct to “…King Abdulaziz City…”

References section. There are typos and inconsistencies in your citation style. You must check them one by one. Some errors:

[6] (extra symbols)

[7] (page is missing)

[16] (use superscript for -2)

[17] (use superscript for 3+)

[20-22] (use subscript for 4)

Authors must read and correct their manuscript

Author Response

Referee 3’s comment

The manuscript titled “Upconversion Nanoparticles for Single Particle Imaging and Quantum Sensing” was revised for its possible publication in Materials MDPI.

I agree with the objectives, analyses, and techniques described by the authors after reading the complete manuscript. As a positive point, I found relevant the results presented. Nevertheless, the manuscript must improve some editing issues and spelling to comply with the quality standards of this journal.

Response: We thank the Referee for his/her valuable time for reviewing, kindly giving us constructive suggestions to improve our manuscript, and positively recommending our manuscript for publication in Nanomaterials.

  • The first keyword is too long; one or up to two words for each keyword are acceptable.

Response:

We have edited the keywords as suggested by reviewer 3. The new changes are highlighted in green in line 26.

  • Page 1. Merged brackets of references 1 to 14 need to be separated.

Response:

   We have fixed this issue in the entire manuscript.

  • Orthography in Section 2.1. Reagents are incorrectly capitalized. Only the first word and after period should be in capital letters.

Response:

We have fixed this issue in section 2.1.

  • Section 2.4. Complete the information of this instrumental and measurement technique.

Response:

We agree with the reviewer in this point, this section became section 2.5 in the revised manuscript. We have added enough experimental details as highlighted in red in lines (113-116).

  •  Figure 5 (a). Use a continuous line in stacked spectra because is difficult to distinguish overlapped intensities.

Response:

We have changed this issue in Figure 5 in the revised manuscript.

  •  Figure 5 (b) and (c). Include error bars for each data point and add slope in Figure 5 (c).

Response:

We have added the scale bars and slopes in Figure 5 in the revised manuscript.

  • Acknowledgments have a misspelling, correct to “…King Abdulaziz City…”.

Response:

We have fixed this typo in the revised manuscript.

  • References section. There are typos and inconsistencies in your citation style. You must check them one by one. Some errors:

[6] (extra symbols)

[7] (page is missing)

[16] (use superscript for -2)

[17] (use superscript for 3+)

[20-22] (use subscript for 4)

Response:

We have fixed all these typos in the revised manuscript.

Round 2

Reviewer 2 Report

The revised MS can be accepted as is. 

The revised MS diserves acceptance.